# Femtosecond exciton dynamics in WSe$_2$ optical waveguides

Aaron J. Sternbach[1✉], Simone Latini[2], Sanghoon Chae[3], Hannes Hübener[2], Umberto De Giovannini[2], Yinming Shao[1], Lin Xiong[1], Zhiyuan Sun[1], Norman Shi[1], Peter Kissin[4], Guang-Xin Ni[1], Daniel Rhodes[3], Brian Kim[3], Nanfang Yu[1], Andrew J. Millis[1], Michael M. Fogler[4], Peter J. Schuck[3], Michal Lipson[5], X.-Y. Zhu[6], James Hone[3], Richard D. Averitt[4], Angel Rubio[2,7] & D. N. Basov[1]

Van-der Waals (vdW) atomically layered crystals can act as optical waveguides over a broad range of the electromagnetic spectrum ranging from Terahertz to visible. Unlike common Si-based waveguides, vdW semiconductors host strong excitonic resonances that may be controlled using non-thermal stimuli including electrostatic gating and photoexcitation. Here, we utilize waveguide modes to examine photo-induced changes of excitons in the proto-typical vdW semiconductor, WSe$_2$, prompted by femtosecond light pulses. Using time-resolved scanning near-field optical microscopy we visualize the electric field profiles of waveguide modes in real space and time and extract the temporal evolution of the optical constants following femtosecond photoexcitation. By monitoring the phase velocity of the waveguide modes, we detect incoherent A-exciton bleaching along with a coherent optical Stark shift in WSe$_2$.

[1] Department of Physics, Columbia University, New York, NY 10027, USA. [2] Max Planck Institute for the Structure and Dynamics of Matter, Luruper Chaussee 149, 22761 Hamburg, Germany. [3] Department of Mechanical Engineering, Columbia University, New York, NY 10027, USA. [4] Department of Physics, University of California, San Diego, La Jolla 92093 CA, USA. [5] Department of Electrical Engineering, Columbia University, New York, NY 10027, USA. [6] Department of Chemistry, Columbia University, New York, NY 10027, USA. [7] Center for Computational Quantum Physics (CCQ), Flatiron Institute, 162 Fifth Avenue, New York, NY 10010, USA. ✉email: as5049@columbia.edu

In the steady-state, it is customary to characterize the behavior of optical waveguide modes with the in-plane component of the complex wavevector, $q_r = q_{1,r} + i q_{2,r}$[1–5]; the subscript r stands for "reference". The real component of the wavevector, $q_{1,r}$, describes the phase velocity of the waveguide mode; the imaginary component of the wavevector, $q_{2,r}$, characterizes attenuation. Previous works have demonstrated that the $q_r$ formalism applied to scanning near-field microscopy (SNOM) visualization of the steady state waveguide modes is well suited to evaluate the anisotropic dielectric tensor of vdW semiconductors[4]. This class of materials hosts uncommonly strong excitonic resonances, which are of particular interest given their substantial impact on the ab-plane optical polarizability in the near-infrared and visible range[3,6–8]. These excitonic resonances yield absorptive and refractive spectral features that are, likewise, encoded in the complex wavevector, $q_r$, of waveguide modes[3]. In this work we focus on the ultrafast dynamics of the extraordinary waveguide mode in the prototypical vdW crystal, WSe$_2$ (Fig. 1a, b) produced by photoexcitation. Photoexcitation perturbs excitons[8–13] prompting changes of the complex wavevector, $q_p$, where the subscript p indicates that $q_p$ is obtained in the "photo-excited" state. Our waveguide imaging data (Figs. 1–3) expose the rich photo-induced response of WSe$_2$ rooted in both coherent and incoherent dynamics of the A-exciton. Coherent dynamics are registered in our WSe$_2$ waveguides at the sub-picosecond (ps) timescale and are attributed to the optical Stark shift[10–12]. Incoherent dynamics persist on longer (10 ps) time scales and are indicative of bleaching of the A-exciton. Our work uncovers the utility of waveguide modes in quantifying non-equilibrium light-driven effects in vdW crystals.

## Results

**Time-resolved nano-imaging of waveguide modes.** To experimentally access and control waveguide modes in vdW crystals we utilize state-of-the-art time-resolved scattering near-field optical microscopy (tr-SNOM)[8,14–18], Fig. 1a. Here, the metalized tip of an atomic force microscope (AFM) is illuminated with p-polarized light. The evanescent field is confined at the tip apex with radius of curvature, $a$. The extreme confinement provides access to local near-fields scattered by the AFM tip with $a = 20$ nm spatial resolution, and waveguides modes with momenta up to $q \sim 1/a$, independent of the wavelength of probe radiation (methods, Supplementary Note 1)[19,20]. Monitoring the amplitude of the near-field scattering signal, $S_r$, (methods) along the surface of the crystal we observe a characteristic periodic pattern (Fig. 1b)[3–5]. A representative line profile in the direction normal to the crystal edge (Fig. 1c–e) highlights periodic oscillations. These imaging data are consistent with the following scenario. The AFM tip launches a waveguide mode, which travels in the bulk of the layered crystals. When the waveguide mode reaches the edge of the crystal after traveling the distance $x$ from the AFM probe, the mode scatters to free space and reaches the detector. By measuring the near-field amplitude as a function of $x$, we visualize the electric field profile of the waveguide mode (see Supplementary Note 1). Non-equilibrium data are obtained by illuminating the crystal with an s-polarized pump beam, with nearly homogeneous intensity across the scanned region (see Supplementary Note 1). In the photo-excited case, the complex wavevector of the waveguided mode is altered. In this work, we focus on the response of WSe$_2$ because its A-exciton resonance lies in the frequency region that can be readily accessed with our tr-SNOM allowing us to fully characterize dynamics of excitons under intense optical pumping of this particular vdW waveguide.

**WSe$_2$ in the steady state.** We begin with the electrodynamics of WSe$_2$ at equilibrium. In Fig. 2a we display line profiles of the near-field amplitude collected with several probe energies, $E$. The experimental profiles exhibit oscillations (Fig. 2a) characterized by the steady-state wavevector, $q_r$. To extract quantitative information from the line profiles we implemented the Fourier Transform (FT) analysis. We generalized the FT procedure to correct the observed wavevector for the angle of incidence of the probe radiation, see Supplementary Note 2. The incidence-corrected FT line profiles reveal peaks located at $q_{1,r}(E)$ that are dependent on the probe energy, $E$ (Fig. 2b). The energy-momentum ($E$-$q_{1,r}$) dispersion of the waveguide mode is plotted in Fig. 2d. A non-monotonic dependence of $q_{1,r}$ on the probe photon energy, $E$ is observed. The prominent 'back-bending' dispersion of $q_{1,r}(E)$ is evident in the vicinity of 1.61 eV where the electromagnetic response is dominated by the A-exciton (Fig. 2c, d; methods). The dispersion of the waveguide mode is uniquely determined by the complex dielectric tensor $\|\varepsilon\|$ of the material whose in-plane component, $\varepsilon_{ab} = \varepsilon_{1,ab} + i\varepsilon_{2,ab}$, is accurately represented with a series of Lorentz oscillators (methods; see Supplementary Fig. 8)[3,21]. The absorption spectra obtained from first principles calculations shares good agreement with our Lorentz model only when excitonic contributions are included (Fig. 2c). The dispersion relationship calculated with the Lorentz model reproduces the experimental results supporting the excitonic interpretation of the dispersion anomaly.

**Femtosecond dynamics in the WSe$_2$ waveguide.** We now introduce a pump beam, as schematically shown in Fig. 1a, to interrogate light-induced effects of WSe$_2$ and their transient dynamics. The photon energy of the pump is chosen to be red-detuned by 50 +/− 8 meV from the A-exciton (methods; see Supplementary Note 5). We fix the energy of the probe 5 meV red-detuned from the center frequency of the A-exciton and explore dynamics of the complex wavevector $q_p$ under pumping. It is instructive to analyze photoinduced effects with the differential wavevector $\delta q_j = q_{j,p} - q_{j,r}$ where the subscript j = 1, 2 refers to the real and imaginary components of the wavevector respectively. Our analysis reveals an abrupt drop of the imaginary part, $q_2$ ($\delta q_2 < 0$) and an equally prompt recovery all occurring within the first picosecond after the photoexcitation (Fig. 3a). In other words, red-detuned photoexcitation transiently suppresses absorption in the vicinity of the excitonic resonance, thus reducing waveguide losses. Suppressed dissipation within the duration of the pump pulse is a hallmark of coherent dynamics. In addition, incoherent photo-induced effects occurring on the picosecond timescale ($\Delta t > 1$ ps) are also evident in the evolution of $q_1$ in Fig. 3b.

We proceed with the inquiry into the non-equilibrium waveguide dispersion by varying the probe photon energy at fixed pump-probe time delay (Fig. 3c–e). We witness anomalies in the dispersion of $\delta q_1$ occurring on the femtosecond timescale when the probe energy is in the vicinity of the A-exciton (Fig. 3e). In order to disentangle various transient mechanisms, we have chosen to describe the differential wavevector data with a model described by an effective value of the in-plane component of the dielectric tensor, $\varepsilon_{ab}$, constructed from a series of Lorentzian oscillators (methods). The validity of this approach is attested by an accurate account of the equilibrium spectra of WSe$_2$ (Fig. 2). To extract changes of the effective value of $\varepsilon_{ab}$, we weakly perturbed the oscillators parameters and obtained good agreement with the pump-probe data. The above analysis reveals that features in the differential dispersion near the A-exciton are caused by two concomitant effects: (i) a 7 meV blueshift of the A-exciton observed only at $\Delta t = 0$; and (ii) bleaching of the A-exciton, defined as spectral broadening and/or a decrease in the oscillator strength (see Fig. 3d). Trend (i) recovers within 1 ps, consistent with the dynamics of suppressed dissipation (Fig. 3a);

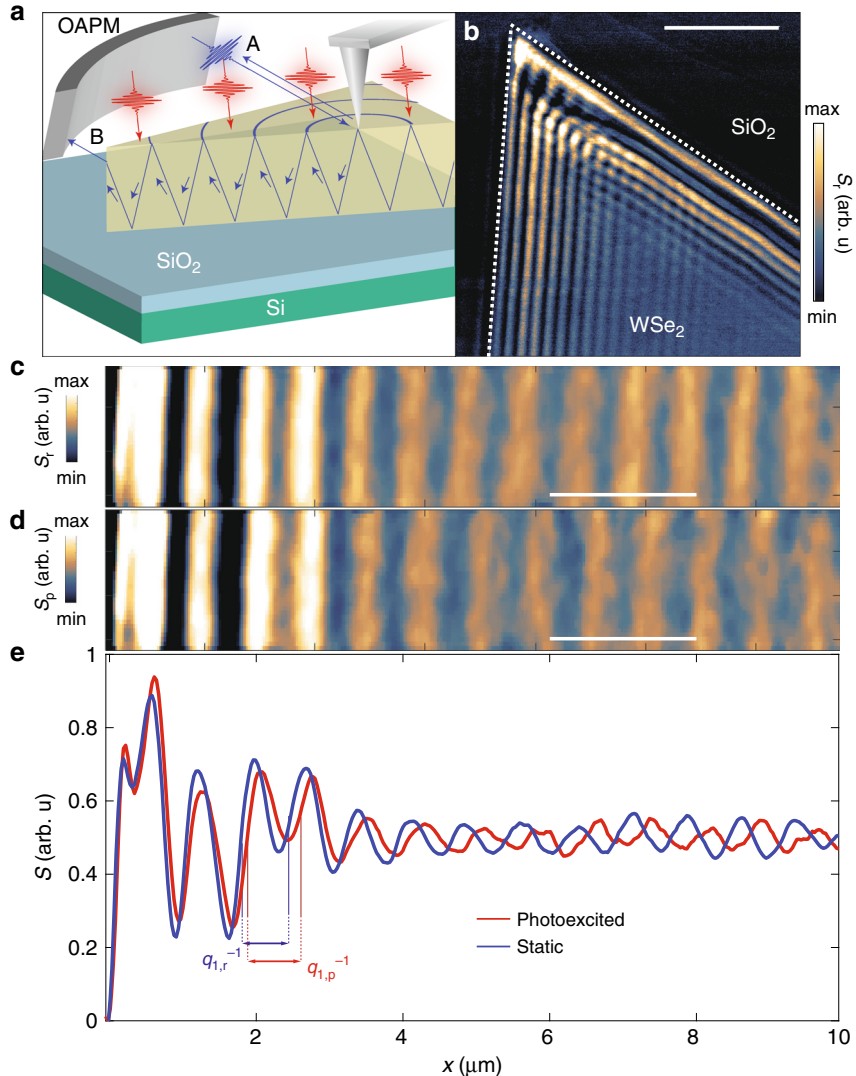

**Fig. 1 Time resolved infrared nano-imaging experiments on WSe$_2$. a** Nano-imaging experiments are performed in a tr-SNOM set-up by shining probe radiation (blue) onto the apex of the AFM tip. Radiation back scattered from the tip-apex (beam A) is collected by an off-axis parabolic mirror (OAPM) and sent to the detector. Simultaneously, a waveguide mode propagating through the crystal slab is launched. Upon reaching the sample edge waveguide modes are scattered to free space (beam B). A second pump channel (red) homogeneously perturbs the sample and alters the propagation of the waveguide modes. **b** Image of the near-field amplitude, $S_r$, at the WSe$_2$/SiO$_2$ interface. The edge of the WSe$_2$ flake was determined from the topography and is indicated by the white dashed line. The scale bar is 5 μm in length. **c** Image of the near-field amplitude collected along the direction normal to an edge of WSe$_2$ in the steady state. **d** Image of the near-field amplitude acquired with sample in its photoexcited state at the time delay of $\Delta t = 1$ ps. A pump with energy 1.58 eV and power 2.5 mW was applied to photo-excite the crystal. The scale bars in panels **c** and **d** are 2 μm in length and the images are co-located. **e** Averaging the 2D data of panels **c** and **d** yields line cuts of the scattering amplitude recorded under equilibrium (blue) and photoexcited (red) conditions, respectively. The experimental traces were vertically offset for clarity. The periodicity of the oscillations is characterized by wavevectors $q_{1,r}$ and $q_{1,p}$ under steady-state and photo-excited conditions respectively. Panels **b-e** were collected with the probe energy, $E = 1.45$ eV on WSe$_2$ crystals with approximately 120 nm thickness.

we therefore conclude that the transient blueshift of the A-exciton resonance is due to a coherent light-induced process[22]. Trend (ii) persists on a much longer timescale and is, therefore, attributed to incoherent exciton dynamics. The dispersion results in Fig. 3 implicate the A-exciton in the observed photo-induced transformations. An overall decrease of the static dielectric function, $\varepsilon_{stat}^*$, which stems from contributions of high-energy excitons outside of our frequency range and also from the response of a photo-excited electron-hole plasma (EHP) is also identified.

## Discussion
The coherent 7 meV blueshift of the A-exciton admits the interpretation within the framework of the optical Stark effect.

The optical Stark shift implies that eigenstates of the system are transiently modified under red-detuned photoexcitation. The Floquet formalism is a customary tool for describing the eigenstates of out-of-equilibrium time-periodic quantum systems[10–12,22,23]. The latter formalism predicts a coherent blueshift of the excitonic resonance, in accord with spectral features we observe in Fig. 3e with zero time-delay between the pump and probe pulses (Table 1; methods). The eigenstates of the hybrid light-matter system were, therefore, found to govern the complex wavevector of the packet of probe photons.

To account for the incoherent response (ii) we consider plausible roles of optically generated carriers. A substantial photo-excited carrier density is independently implied by our observation of transient spectral weight in the mid-infrared (see

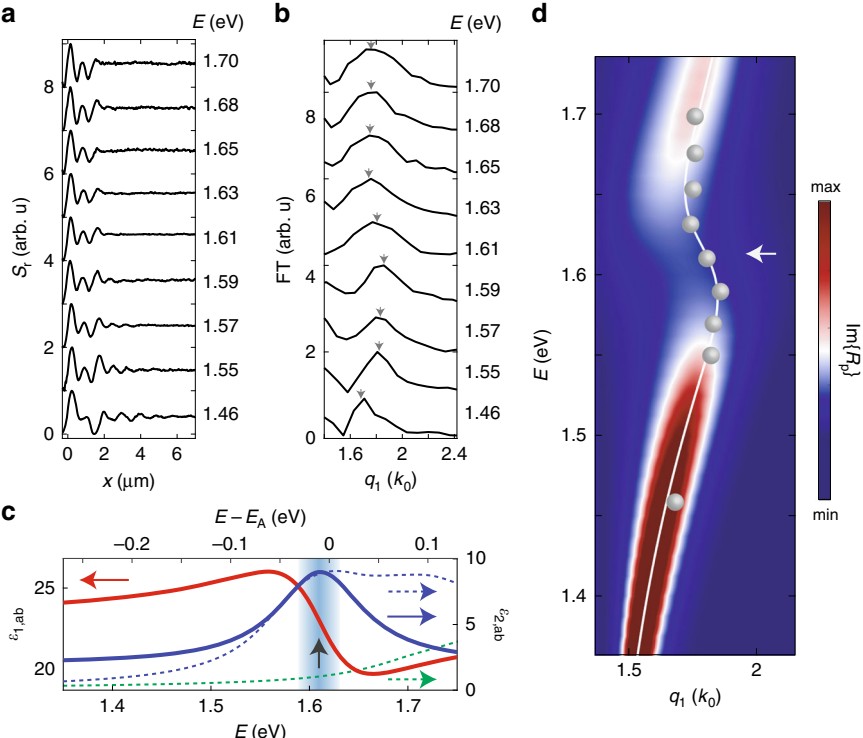

**Fig. 2 Steady-state electrodynamics of the WSe₂ crystal from spatio-temporal nano-imaging. a** Near-field scattering amplitude, $S_r$, at several probe energies (1.46–1.70 eV) plotted as a function of the real space coordinate relative to the edge of the 90 nm thick WSe₂ crystal. **b** Fourier transforms of the data in panel a are shown in units of $k_0$, the wavevector of light in free space, after angular correction (see Supplementary Note 2). The arrows indicate the real part of the dominant wavevector, $q_{1,r}$ (see Supplementary Notes 2–4). The data presented in panels **a** and **b** are offset for clarity. **c** The in-plane component of the dielectric function of bulk WSe₂ ($\varepsilon_{1,ab}$ shown in red, $\varepsilon_{2,ab}$ shown in blue) obtained from a series of Lorentzian oscillators with parameters reported in Supplementary Table 1 (see Supplementary Note 8). The imaginary component of the dielectric function obtained from first principles calculations including excitonic effects (dashed blue curve) can be compared with the result obtained when excitonic effects are excluded (dashed green curve). The energy scale for the theoretical calculations is on the top, which was offset by the A-exciton energy (see Supplementary Note 8). The A-exciton energy, $E_x$, is indicated with a black arrow. **d** The energy versus momentum dispersion of the waveguide mode calculated with the in-plane component of the dielectric function represented by the Lorentz model (shown in panel **c**). The dispersion is represented with the imaginary component of the p-polarized reflection coefficient, Im{$R_p$} (false color map) along with the analytical solution (white line; Supplementary Notes 3, 4). The experimental results for $q_{1,r}$ vs. E extracted from the data of panels a and b are displayed with gray dots. The energy $E_x$ is indicated with a white arrow.

Supplementary Fig. 9), which may be produced by bound excitons and/or a Drude response from itinerant carriers. Bound excitons cannot account for the observed pump induced change of the static dielectric function, $\varepsilon_{stat}^*$, which is caused by optical transitions at higher energies than those investigated in our work. The net carrier density extracted from time-resolved mid-infrared spectroscopy (see Supplementary Fig. 9) is approximately $n_e = 10^{19}$ cm$^{-3}$, which is sufficient to reach the Mott transition threshold[13] provided that a significant fraction of $n_e$ is comprised of an optically generated EHP (see Supplementary Note 9). At the Mott threshold, the EHP screens the Coulomb interaction that binds excitons, which causes excitons to dissociate. As the EHP density approaches the Mott threshold, elastic scattering of excitons leads to spectral broadening of the excitonic transitions while screening of the Coulomb interaction suppresses the excitons' oscillator strengths[9,13,24–26]. Both of these effects are manifest in trend (ii) above. The same physical mechanisms that produce trend (ii) are likely to influence high-energy excitons outside of our frequency range, which would impact $\varepsilon_{stat}^*$. We comment that photoexcitation is also likely to cause both Joule and electronic heating of the WSe₂ crystal, which results in spectral broadening of the A-exciton resonance[27] and may also lead to a decrease of $\varepsilon_{stat}^*$. Heating, therefore, may partially contribute to our observation of trend (ii), but is unrelated to trend (i) occurring on much faster time scales.

In conclusion, transient nano-imaging of waveguide modes allowed us to quantify photo-induced changes of optical properties of WSe₂ with sub-ps temporal resolution. The procedures detailed here are expected to remain valid provided that the effective optical constants do not change appreciably during the course of waveguide propagation. If the latter condition is not satisfied then uncommon phenomena, including light amplification without population inversion, could be observed[28]. Our analysis reveals that the magnitude of the pump-induced changes of the effective in-plane component of the dielectric function, $\delta\varepsilon_{1,ab}$(E = 1.45 eV) reaches 1.3 and therefore exceeds 5% of the value of $\varepsilon_{1,ab}$ at equilibrium (Fig. 1). Substantial photo-induced changes of the phase velocity were also observed on WS₂ indicating that the observed effects are likely generic within vdW semiconductors (see Supplementary Note 13). Our observations, furthermore, suggest the possibility to substantially tune the optical birefringence of vdW semiconductors on-demand (see Supplementary Note 10). The spectral analysis of $\delta q_1(E)$ presented in Fig. 3, which are recast as $\delta\varepsilon_{1,ab}(E)$ in Supplementary Fig. 10, indicate that the observed changes of the dielectric function are dominated by photo-induced perturbation of the A-exciton resonance. Both coherent and incoherent light-induced processes were revealed through systematic study of the dynamics of the complex wavevector of the elliptic waveguide modes. Our work expands on pioneering efforts to monitor ultrafast exciton dynamics with tr-

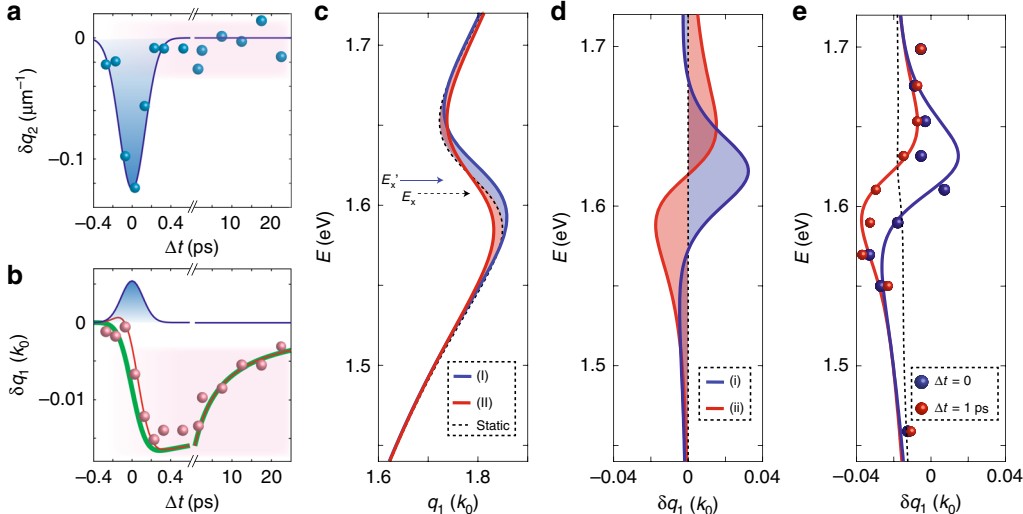

**Fig. 3 Differential dispersion and dynamics of the waveguide mode in WSe$_2$.** These data were collected on a 90 nm slab of WSe$_2$ with the probe energy $E = 1.61$ eV and pump energy $E = 1.56$ eV (Supplementary Notes 5 and 7). **a**, **b** were collected with a pump power of 1.5 mW (see Supplementary Note 7 for fluence dependence). **a** Values of $\delta q_2$ (blue dots) are displayed as a function of $\Delta t$. The solid blue line indicates the pump-probe convolution. **b** Values of $\delta q_1$ (red dots) are extracted from the same dataset used in panel **a**. The solid green line is calculated from an Auger model (see Supplementary Note 9), while the blue line indicates the pump-probe convolution. The red line is the sum of the latter two functions. **c**, Lorentz model calculations of the dispersion relationship in equilibrium and non-equilibrium conditions (Table 1; Supplementary Note 8). The calculations show the equilibrium dispersion relationship (dashed black line) in the vicinity of the A-exciton, initially at energy $E_x$, (black dashed arrow). A non-equilibrium blueshift of the A-exciton[10] to an energy $E_x' = E_x + 7$ meV (blue arrow) produces trend (I) (blue line). Bleaching of the A-exciton (see main text) produces trend (II) (red line). **d** The differential dispersion defined as the trend of $\delta q_1 = q_{1,p} - q_{1,r}$ vs. $\omega$. The differential dispersion of the blue-shifted A-exciton[10] produces the trend (i) (blue line). The differential dispersion of the bleached A-exciton produces the trend (ii) (red line), see main text. **e** The experimental differential dispersions were obtained at two-time delays $\Delta t = 0$ (blue dots) and $\Delta t = 1$ ps (red dots) with a pump power of 3 mW. The red and blue lines are calculations using the parameters reported in Table 1. Bleaching of the A-exciton, trend (ii), is prominent in $\Delta t = 1$ ps data (red line), while a blue-shift of the A-exciton, trend (i), plays a dominant role in $\Delta t = 0$ data (blue line). An approximately linear decrease of $q_1(E)$, produced by the decrease of $\varepsilon^*_{stat}$, is also included in the calculations at both time delays (black dashed line).

**Table 1 Parameters of the Lorentz model for the dielectric function.**

|  | $\delta\varepsilon_{stat}$ | $\delta N$ | $\delta E$ (meV) |
|---|---|---|---|
| Equilibrium | 0 | 0 | 0 |
| $\Delta t = 1$ ps | 0.6 | 0.12 | −0.5 |
| $\Delta t = 0$ | 0.5 | 0.09 | 7 |

SNOM[8] and paves the way to utilize waveguide modes as a quantitative spectroscopic tool to interrogate light-induced phases at the nanoscale. The inherent spatial resolution of this pump-probe method may be suitable for imaging edge modes and edge effects in topologically nontrivial materials[29].

## Methods

**Experimental set-up.** For nano-optical imaging experiments, we used a scattering-type scanning near-field optical microscope (s-SNOM, Neaspec). The atomic force microscope (AFM) operates in tapping-mode with a frequency of about 70 kHz and a tapping amplitude of approximately 50 nm. The time-resolved Pseudoheterodyne technique is used to extract the amplitude ($S$) and phase ($\psi$) of the near-field signal[14]. In this work we discussed the amplitude, $S$. The additive background contribution was strongly attenuated by considering only information modulated at the third harmonic of the tapping frequency of the AFM tip. The pump and probe channels were derived from a 20 W, 1030 nm Yb:kGW amplified laser source operating at a repetition rate of RR = 750 kHz (Pharos, Light Conversion). The intense laser pulse was converted to two separate, pump and probe channels of visible radiation by employing two optical parametric amplifiers (OPAs). To facilitate self-referencing as described below, and in detail within Supplementary Note 1 and ref. [14], the pump pulse operates at half of the repetition rate, RR/2 which is approximately 375 kHz. The broadband pulses were sent through band-pass filters (approximately 15-20 meV FWHM) before they were delivered to the Neaspec microscope (see Supplementary Note 12). For all data displayed in main text the center energy of the pump was fixed at approximately 1.56 eV.

We collect the self-referenced and time-resolved near-field amplitude[14] (see Supplementary Note 1). Specifically, we acquire the near-field amplitude in the steady-state ($S_r$) as well as in the photo-excited state ($S_p$) 'simultaneously' at each pixel using homebuilt software as described in Supplementary Note 1 and in ref. [14]. The differential change in the near-field amplitude is calculated as $\Delta S = (S_p - S_r)/\langle S_r \rangle$ where $\langle S_r \rangle$ is the mean value of the reference near-field amplitude in the interior of WSe$_2$. The quantity $\Delta S$ is used to extract differential changes in the complex wavevector as we describe in the Supplementary Notes 1, 2, 4 and 5. The samples studied here are WSe$_2$ planar waveguides. The high quality WSe$_2$ crystals were synthesized by self-flux method, then mechanically exfoliated onto an SiO$_2$/Si substrate[30].

**Model for the dielectric function of WSe$_2$.** The dielectric tensor of WSe$_2$ determines the complex wavevector of waveguide modes that propagate within these crystals as described in Supplementary Note 3 and in refs. [3,4]. A Lorentz model is used to approximately describe the in-plane component of the dielectric function of WSe$_2$:

$$\varepsilon_{ab} \cong \varepsilon^*_{stat} - \delta\varepsilon_{stat} + \frac{f_1(1 - \delta N \cdot C)}{(E_1 + \delta E)^2 - E^2 - iE\gamma_1(1 + \delta N \cdot (1 - C))} \tag{1}$$

In this expression the parameters $f_1, \gamma_1, E_1$ represent, respectively, the oscillator strength, spectral breadth and center frequency of the A-exciton respectively at equilibrium. The parameter $\varepsilon^*_{stat}$ represents the static dielectric function, which absorbs contributions from high-energy (B, C, D, etc.) excitons as well as contributions from interband optical transitions (see Supplementary Note 8). The parameters are given by $f_1 = 1.2$ eV$^2$, $\gamma_1 = 106$ meV, $E_1 = 1.612$ eV, and $\varepsilon^*_{stat} = 22.5 + 2i$ as described in Supplementary Note 8 and describe the in-plane component of the dielectric function in the steady state. The remaining parameters in Eq. (1) represent differential changes to the dielectric function in the photo-excited situation. The parameter is $\delta\varepsilon_{stat}$ represents a change in the static dielectric function. The parameter $\delta E$ represents a shift of the center frequency of the A-excitons while $\delta N$ represents a change in the oscillator strength and/or spectral breadth. Finally, $C$ determines the relative value of the change in oscillator strength vs. a change of the spectral breadth. An arbitrary choice of this parameter on the interval [0,1] provides an adequate fit to our experimental data. Values for these parameters, that are extracted from the measurements reported in Fig. 3, are given in Table 1. The out-of-plane dielectric function is dispersion-less within the

experimentally investigated range of probe energies. Thickness dependence of the complex wavevector in the steady state (see Supplementary Note 11) are used to extract the out-of-plane component of the dielectric function given by a constant value of $\varepsilon_c = 8$ that is assumed to be unchanged by photoexcitation. We emphasize that the out-of-plane optical polarizability exhibits negligible dispersion in the steady state[31]. Thus, possible pump-induced changes of $\varepsilon_c$ would have a negligible impact the extracted values of $\delta N$ or $\delta E$, which give rise to the dispersive trends observed in Fig. 3e.

## Data availability
The data that support the plots within this paper and other findings of this study are available from the corresponding author upon reasonable request.

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

## Acknowledgements
This work is supported as part of Programmable Quantum Materials, an Energy Frontier Research Center funded by the U.S. Department of Energy (DOE), Office of Science, Basic Energy Sciences (BES), under award DE-SC0019443. D.N.B. is the Vannevar Bush Faculty Fellow (N00014-19-1-2630). We acknowledge support from the European Research Council (ERC-2015-AdG694097) and the Cluster of Excellence 'Advanced Imaging of Matter' (AIM). Support by the Flatiron Institute, a division of the Simons Foundation, is acknowledged. S. L. acknowledges support from the Alexander von Humboldt foundation.

## Author contributions
A.J.S. and S.H.C., conducted the nanoscale infrared measurements and sample characterization. A.J.S., P.K., R.D.A. and D.N.B. contributed to the development of scientific instrumentation used in this study. S.H.C., D.R., B.K. and J.H. designed and created samples and devices used in this work. S.L., H.H., U.D.G., L.X., Z.S., A.J.M., M.M.F. and A.R. provided theoretical support. A.J.S., N.S. and N.Y. performed far-field optical characterizations. Y.S., L.X., G.X.N., P.J.S., M.L. and X.Y.Z. helped to interpret the results. D.N.B. supervised the project. A.J.S., S.L., A.R. and D.N.B. co-wrote the manuscript with input from all co-authors.

## Competing interests
The authors declare no competing interests.
