## [Peer Review File · Nature Communications]

REVIEWERS' COMMENTS:

Reviewer #1 (Remarks to the Author):

The authors did an excellent job - they answered all my questions and improve the presentation of their work considerably. I recommend the publication of the manuscript in the journal of Nature Communications.

The only thing that is "wrong" is the formula for group velocity. If we look at the Figure 2d and use this formula ($dE/dq1r$) we would obtain infinite value of group velocity at the edges of anomalous dispersion (and negative value in the region of anomalous dispersion) - which is obviously wrong - group velocity should describe the speed of transfer of energy and should be smaller than the speed of light. It is well-known that the formula used in the manuscript is valid only for small dissipation. There is a tradition of using this formula not only in the transparency window but well outside its validity - but I am not sure that this tradition is justified. I leave to authors whether they would like to address this point (as it basically depends on definitions).

Reviewer #2 (Remarks to the Author):

Dear Editor and Authors,

I think the authors provided satisfactory answers to my comments and also implemented appropriate changes to the revised manuscript addressing the questions.

Reviewer #3 (Remarks to the Author):

After reading the authors' response for all the reviewers' comments, I find that they have resolved the majority of the concerns that were raised by the reviewers - the majority of the questions/points/issues were indeed fully clarified by the authors, yet in some of the cases (especially in responding to the remarks of reviewer 1), the answers are a bit laconic.

Nevertheless, I feel that the current manuscript (with the detailed SOM) is well written, very interesting, and brings new knowledge to the field, thus I support publication in your Nat. Comm.

We would like to thank all reviewers for considering our manuscript. The remaining concerns of the reviewers have been addressed, as described in the point-by-point response below.

Response to Reviewer 1:

The authors did an excellent job - they answered all my questions and improve the presentation of their work considerably. I recommend the publication of the manuscript in the journal of Nature Communications.

The only thing that is "wrong" is the formula for group velocity. If we look at the Figure 2d and use this formula (dE/dq_{1r}) we would obtain infinite value of group velocity at the edges of anomalous dispersion (and negative value in the region of anomalous dispersion) - which is obviously wrong - group velocity should describe the speed of transfer of energy and should be smaller than the speed of light. It is well-known that the formula used in the manuscript is valid only for small dissipation. There is a tradition of using this formula not only in the transparency window but well outside its validity - but I am not sure that this tradition is justified. I leave to authors whether they would like to address this point (as it basically depends on definitions).

We thank reviewer #1 for raising our attention to this issue and for the recommendation to publish our article. We have replaced the sentence under question with: "A non-monotonic dependence of $q_{1,r}$ on the probe photon energy, E is observed. A prominent back-bending in the dispersion of $q_{1,r}(E)$ is evident in the vicinity of 1.61 eV where the electromagnetic response is dominated by the A-exciton (Fig. 2c,d; methods)."

Response to reviewer #2:

I think the authors provided satisfactory answers to my comments and also implemented appropriate changes to the revised manuscript addressing the questions.

We thank reviewer #2 for providing helpful comments and for the positive review.

Response to reviewer #3:

After reading the authors' response for all the reviewers' comments, I find that they have resolved the majority of the concerns that were raised by the reviewers - the majority of the questions/points/issues were indeed fully clarified by the authors, yet in some of the cases (especially in responding to the remarks of reviewer 1), the answers are a bit laconic.

Nevertheless, I feel that the current manuscript (with the detailed SOM) is well written, very interesting, and brings new knowledge to the field, thus I support publication in

your Nat. Comm.

We thank reviewer #3 for helpful comments and for recommending our article for publication.